# microRNA Expression Dynamics in *Culicoides sonorensis* Biting Midges Following Blood-Feeding

**DOI:** 10.3390/insects14070611

**Published:** 2023-07-06

**Authors:** Mary Katherine Mills, Paula Rozo-Lopez, William Bart Bryant, Barbara S. Drolet

**Affiliations:** 1Department of Biology and Geology, University of South Carolina-Aiken, Aiken, SC 29801, USA; 2Department of Microbiology, University of Tennessee Knoxville, Knoxville, TN 37996, USA; plopez2@utk.edu; 3Department of Medicine and Vascular Biology Center, Medical College of Georgia, Augusta University, Augusta, GA 30912, USA; wilbryant@augusta.edu; 4Arthropod-Borne Animal Diseases Research Unit, Center for Grain and Animal Health Research, Agricultural Research Service, United States Department of Agriculture, Manhattan, KS 66502, USA; barbara.drolet@usda.gov

**Keywords:** miRNA, ncRNA, vector, *Culicoides*, transcriptome, biting midges, blood feeding

## Abstract

**Simple Summary:**

*Culicoides sonorensis* midges are able to transmit pathogenic viruses to livestock, resulting in significant economic losses. Virus transmission is linked to blood feeding, where female *C. sonorensis* midges require blood to produce eggs. Due to the relationship between egg development, blood feeding, and pathogen transmission, there is a need to understand midge physiology to control virus transmission. One aspect of midge physiology that has yet to be explored is microRNAs (miRNAs), which display different expression patterns related to egg development and pathogen infection. miRNAs degrade messenger RNAs in a sequence-specific manner which stops associated protein production. To determine the *Culicoides* miRNA catalog, we sequenced small RNA molecules (small RNA-Seq) and used bioinformatic analyses (miRDeep2) of whole female midges and digestive tissues before and after blood feeding. Our analyses characterized 76 miRNAs within *C. sonorensis*. Based on our findings, we suggest an interesting evolutionary relationship between miRNA expression and blood meal requirements across blood-feeding insects. We also identified miRNAs with expression patterns regulated by blood meal ingestion and/or tissue. Lastly, we identified potential miRNAs regulated by virus infection. Overall, our data provide a foundation for future studies to determine the mechanisms of gene regulation associated with midge physiology and virus transmission.

**Abstract:**

*Culicoides sonorensis* midges vector multiple livestock arboviruses, resulting in significant economic losses worldwide. Due to the tight association between virus transmission, blood feeding, and egg development, understanding midge physiology is paramount to limiting pathogen transmission. Previous studies have demonstrated the importance of small non-coding RNAs (ncRNAs), specifically microRNAs (miRNAs), in multiple aspects of vector physiology. These small ncRNAs regulate gene expression at the post-transcriptional level and display differential expression during pathogen infection. Due to the lack of annotated miRNAs in the biting midge and associated expression profiles, we used small RNA-Seq and miRDeep2 analyses to determine the *Culicoides* miRNAs in whole females and midgut tissues in response to blood feeding. Our analyses revealed 76 miRNAs within *C. sonorensis* composed of 73 orthologous and three candidate novel miRNAs, as well as conserved miRNA clusters. miRNA conservation suggests an interesting evolutionary relationship between miRNA expression and hematophagy in the infraorder Culicomorpha. We also identified multiple blood meal-regulated and tissue-enriched miRNAs. Lastly, we further identified miRNAs with expression patterns potentially associated with virus infection by probing publicly available datasets. Together, our data provide a foundation for future ncRNA work to untangle the dynamics of gene regulation associated with midge physiology.

## 1. Introduction

*Culicoides* biting midges (Diptera: Ceratopogonidae) are small hematophagous flies that vector pathogenic viruses to susceptible livestock populations, including bluetongue virus (BTV), Schmallenberg virus (SBV), epizootic hemorrhagic disease virus (EHDV), and vesicular stomatitis virus (VSV), which can result in severe economic losses due to increased livestock mortality, morbidity, production loss, and trade restrictions [1,2,3]. *Culicoides* females transmit pathogenic viruses through repeated blood feeding, which is a physiological requirement to produce eggs [3,4]. Due to the tight association between pathogen transmission, blood feeding, and egg development, understanding midge physiology is critical to developing more targeted control strategies and reducing pathogen transmission.

In addition to molecular and hormonal cascades, vector physiology and immune responses are regulated in part by small non-coding RNAs (ncRNAs). These RNA molecules are less than 200 nucleotides (nt) in length that post-transcriptionally regulate gene expression [5]. MicroRNAs (miRNAs), one group of small ncRNAs, are 22 nt in length that post-transcriptionally regulate gene expression by binding to complementary messenger RNA (mRNA) targets [6]. This group of small ncRNAs can reside in the genome individually or in clusters [7,8] and is involved in tissue homeostasis and immunity in multiple organisms [9,10,11,12]. In mosquitoes, miRNAs display various expression patterns, including tissue enrichment, tissue exclusion, pan-tissue expression, and differential expression after stimuli, such as blood feeding or pathogen infection [13,14,15,16,17,18,19,20,21,22,23,24,25,26,27]. These expression patterns suggest miRNAs may also play a role in arboviral infection progression. Despite the importance of miRNAs, there were no annotated miRNAs available for *C. sonorensis*, emphasizing the need to explore small ncRNA dynamics within this understudied vector.

Although the role of miRNAs in midge physiology remains elusive, many aspects of *C. sonorensis* vector competence have been determined. Infection time course studies have identified the midgut as the primary barrier to arboviral infection [28,29,30,31]. For successful arbovirus transmission, the pathogen must first establish infection in the midgut epithelium after ingestion. To reach secondary tissues (i.e., salivary glands), virions must pass the midgut escape barrier and disseminate through the midgut basal lamina into the hemolymph. BTV, EHDV, and VSV dissemination into the hemolymph has been recorded as early as 72 h post-ingestion of an infectious blood meal [28,29,30,32], suggesting that virus–midgut interactions occur within the first 72 h of infection. For the subsequent bite transmission, virions must establish infection within the salivary glands to be released during subsequent blood feeding [31,33,34]. Unlike some mosquito vector species, the salivary glands of *Culicoides* do not serve as significant barriers to infection [29]. Therefore, investigating the physiology of the midgut–virus interface is critical to understanding vector competence, and to the identification of targets to limit pathogen transmission by *Culicoides* midges.

We took a deep sequencing approach to generate a comprehensive miRNA catalog of *C. sonorensis*. Through our analyses, we identified potential *Culicoides*-specific miRNAs and multiple miRNA clusters associated with hematophagy. We also determined the miRNA expression dynamics of whole females and midgut tissues before and after blood feeding. Furthermore, we analyzed publicly available small RNA datasets [35] to identify miRNAs with infection-correlated expression patterns. Overall, our data provide an initial, thorough *Culicoides* miRNA catalog, which is a necessary foundation for future miRNA studies in biting midges.

## 2. Materials and Methods

### 2.1. Culicoides sonorensis Maintenance and Blood Feeding

Colonized *C. sonorensis* adult midges (AK colony, USDA, Arthropod-Borne Animal Diseases Research Unit, Manhattan, KS, USA) were used for all experiments. Midges were maintained in environmental chambers at 25 ± 0.5 °C with 70 ± 5% R.H. and 13:11 light/dark cycle and offered 10% sucrose solution ad libitum. Newly emerged females (1–3 days post-emergence) were blood-fed on a 1:1 mixture of defibrinated sheep blood (Lampire Biological Products, Pipersville, PA, USA) and Eagle’s minimum essential medium (Sigma, St. Louis, MO, USA). Midges were allowed to feed for 60 min on glass bell jars connected to a water-jacketed system warmed at 37 °C with a parafilm membrane/cage interface. After blood feeding, midges were anesthetized with CO_2_, and fully engorged females were selected, placed in cardboard maintenance cages, and fed 10% sucrose ad libitum. A second group of age-matched females was maintained on 10% sucrose solution only to be used as teneral, sugar-fed, controls.

### 2.2. Sample Collection

#### 2.2.1. Insect Sample Collection

All samples were collected in 500 µL TRIzol (Invitrogen; Thermo Fisher Scientific, Inc., Waltham, MA, USA) from three biological replicates (Appendix A) and stored at −80 °C until further processing. To perform RNA-Seq, whole females (*n* = 20 per treatment per replicate) and midgut-enriched digestive tracts (*n* = 100 per treatment per replicate) were collected 72 hpbm and from aged-matched sugar-fed midges. To confirm the blood meal-regulated expression profiles of *cso-miR-2944a* and *cso-miR-286a*, whole blood-fed (24, 48, and 72 hpbm) and aged-matched sugar-fed females (*n* = 20 per treatment/time point per replicate) were collected. To confirm the tissue-enriched expression profile of *cso-miR-1174,* midgut-enriched digestive tracts and the remaining tissues of the dissected bodies from sugar-fed females (*n* = 30 per replicate) were collected 72 h post-emergence.

#### 2.2.2. Cell Line Sample Collection

Uninfected *Culicoides* cell line ABADRL-Cs-W8A was maintained as previously described [36]. A fully confluent T-25 flask of W8A cells was collected, with three separate flasks used for three biological replicates.

### 2.3. RNA Isolation

Frozen TRIzol insect samples were thawed on ice. Insect samples were homogenized using a Bead Mill Homogenizer (Omni Inc., Kennesaw, GA, USA) set at 3.1 m/s for 2 min with two 2.4 mm stainless steel beads (Omni Inc.) per sample. Cells were lysed by adding 1 mL of TRlzol and homogenized by pipetting the suspension several times. Total RNA was extracted using a final volume of 1 mL TRIzol in TRIzol Phasemaker Gel Tubes (Invitrogen; Thermo Fisher Scientific, Inc., Waltham, MA, USA) according to the manufacturer’s recommendations. Pellets were air-dried and resuspended in 40 μL RNAse-free water (Fisher Scientific, Waltham, MA, USA). RNA concentration was determined by Nanodrop (Thermo Fisher Scientific, Waltham, MA, USA). Only samples with 260:280 and 260:230 ratios between 1.8 and 2.0 were used for further analyses.

### 2.4. Sequencing

RNA extracted from insect and W8A cell samples (Appendix A) was sent to the University of Kansas Medical Center Genomics Core for library preparation and illumina sequencing. For each tissue sample, RNA quality was assessed with Agilent 2100 Bioanalyzer using the RNA6000 Nano kit chip (Agilent, Santa Clara, CA, USA). For Illumina library construction, 1ug of total RNA was used to initiate the TruSeq Small RNA library preparation protocol (Illumina, San Diego, CA, USA). The total RNA was ligated with 3′ and 5′ RNA adapters followed by a reverse transcription reaction and a 15 cycle PCR amplification, which incorporates the small RNA TruSeq index adaptor. Size selection of the cDNA library construct was conducted using 3% marker F gel cassettes—dye-free with internal standards on the Pippin Prep size fractionation system (Sage Science) using a 125 bp–160 bp size capture. Size-captured library constructs were purified using Qiagen MinElute PCR purifications kit (Qiagen, Hilden, Germany). The Agilent TapeStation 4200 was used with the High Sensitivity DNA1000 ScreenTape assay (Agilent) to validate the purified libraries. Following Agilent TapeStation QC of the library preparation and library quantification using the Roche Lightcycler96 with FastStart Essential DNA Green Master (Roche, Indianapolis, IN, USA), the RNA-Seq libraries were adjusted to a 2 nM concentration and pooled for multiplexed sequencing. Libraries were denatured and diluted to the 2 pM concentration (based on qPCR results) followed by clonal clustering on an Illumina NextSeq 550 using a Mid-Output v2.5 reagent kit (Illumina) with a 1 × 50 bp cycle sequencing profile with a single index read. Following collection, sequence data were converted from .bcl file format to fastq files and de-multiplexed into individual sample sequence datasets for further downstream analysis. The 12 libraries consisted of five groups: sugar-fed (SF) whole body (WB), blood-fed (BF) WB, SF midgut (MG), BF MG, and W8 with three biological replicates per group.

### 2.5. Bioinformatic Analyses

FASTQ files from the 15 small RNA-Seq libraries were analyzed using CLC Genomics Workbench v21 (Qiagen). Raw reads were trimmed of adaptor sequences and mapped to the *Culicoides sonorensis* PIR-s-3 genome Cson1.2 [37] from Vectorbase [38] (Appendix A).

### 2.5.1. miRNA Identification

Unique *Culicoides*-mapped reads were queried against *Aedes aegypti* precursor sequences from miRBase v22 [39,40,41,42,43,44], allowing for a 2 nt mismatch. *Culicoides*-mapped reads matching *Ae. Aegypti* precursor sequences were consolidated by group and fed into miRDeep2 using default parameters [45]. miRDeep2-generated precursor sequences with (1) ≥1 star read, (2) significant randfold *p*-values, and (3) >200 total read count were considered orthologous miRNAs and used in further analyses. *Culicoides*-mapped reads not matching *Ae. Aegypti* precursor sequences were also consolidated by group and fed into miRDeep2 using default parameters. miRDeep2-generated precursor sequences with (1) ≥1 star read, (2) significant randfold *p*-values, and (3) >200 total read count were manually queried against miRBase v22 to identify additional orthologous miRNAs with >2 nt mismatches. Non-orthologous, miRDeep2-generated precursor sequences were fed into RNAfold [46] to determine candidate novel *Culicoides* miRNAs. Resulting sequences with (1) a 2 nt 3′ overhang [47], (2) <−20 kcal/mol MFE [48], and (3) expression profiles manually verified from *Culicoides*-mapped reads in CLC Genomics Workbench were considered candidate novel *Culicoides* miRNAs. Using the genome location of *Culicoides* miRNAs, miRNA clusters were identified as miRNAs residing within 10,000 bp of each other [7,8]. miRNAs *cso-miR-219, cso-miR-2944b, cso-miR-2946, cso-miR-309,* and *cso-miR-971* have <200 total read count and were not assessed using differential gene expression analyses but were considered as orthologous miRNAs due to miRDeep2 analyses and manual assessment of mapped reads. All *Culicoides* miRNAs are available in Appendix A, which contains the miRNA precursor sequence, mature sequence, associated sequence lengths, genome location, and clustering. *Culicoides* miRNA precursor and mature sequences are available in FASTA format in Appendix A, respectively. PCA was performed on insect sample reads mapping to *Culicoides* miRNA precursor sequences across treatment (blood meal) and tissue (Appendix A).

#### 2.5.2. Identification of Conserved miRNAs

To identify conserved miRNAs across insect taxa, *Culicoides* mature and precursor miRNA sequences were queried against Diptera (*Aedes aegypti*, *Anopheles gambiae*, *Culex quinquefasciatus*, *Drosophila melanogaster*, and *Drosophila mojavensis*), Lepidoptera (*Bombyx mori*, *Heliconius melpomene*, and *Manduca sexta*), Hymenoptera (*Apis mellifera*), Coleoptera (*Tribolium castaneum*), and Hemiptera (*Acyrthosiphon pisum*) miRNAs using the miRBase v22 BLAST analyses (Appendix A). Resulting miRNAs with e values < 0.005 were considered conserved as previously described [49].

#### 2.5.3. Differential Expression Analyses

The consolidated list of orthologous and candidate novel *Culicoides* miRNA precursor sequences (Appendix A) with >200 total read counts were fed into RNA-Seq and differential expression analysis using default parameters in CLC Genomics workbench. Resulting differential expression data were used to identify miRNA expression enrichment across blood meal and tissue in all insect samples. miRNAs were considered to have an enriched expression when there was a statistically significant ≥ 2-fold change between the treatment (BF v SF) or tissue (WB v MG) (Table 1 and Appendix A). miRNAs with enriched expression between blood meal or tissue were cross-referenced to identify miRNAs with blood meal-regulated and tissue-enriched expression (Table 1 and Appendix A). miRNAs with expression enrichment associated with whole body over midgut tissues were termed enriched in non-midgut tissues. Such tissues include ovaries, fat body, salivary glands, neuronal, and muscular tissues. Normalized log_10_
*Culicoides* miRNA expression values (Appendix A) were fed into Morpheus (Broad Institute, Cambridge, MA, USA; software.broadinstitute.org/morpheus/ (accessed on 3 April 2023)) to generate a heatmap (Figure 1) as previously described [15].

#### 2.5.4. Identification of miRNAs Potentially Involved in Viral Infection

To identify potential *Culicoides* miRNAs involved in viral infection, publicly available small RNA datasets from the European Nucleotide Archive (ENA) accession number ERP001936 were retrieved [35]. These datasets represent small RNA transcriptomes from *Culicoides* KC cells exposed to either Schmallenberg virus (KC SBV 1: ERR186232; KC SBV 2: ERR186233) or bluetongue virus serotype 1 (KC BTV 1: ERR186226; KC BTV 2: ERR186227). FASTQ files were retrieved, trimmed of adaptors, and mapped to the *C. sonorensis* PIR-s-3 genome Cson1.2 using CLC Genomics Workbench v21. To determine miRNA expression values for infected samples, *Culicoides* miRNA precursor sequences (Appendix A, Appendix A) were fed into RNA-Seq and miRDeep2 using default parameters. The resulting data were compiled with expression data from insect and W8A cell samples to identify the ten highest and lowest expressed miRNAs (Appendix A) within each dataset.

### 2.6. RT-qPCR Analyses

To validate small RNA-Seq expression profiles using RT-qPCR, 20 ng of total RNA was reverse transcribed in a 20 μL volume reaction using the miRCURY LNA RT Kit (QIAGEN, Hilden, Germany). RT-qPCR was carried out with the miRCURY LNA SYBR Green PCR Kit (QIAGEN) and miR-specific LNA primers (miRCURY LNA miRNA Custom PCR Assay, QIAGEN; Appendix A) using ROX reference dye as a 200× concentrate according to the AB7500 system fast mode protocol. PCR cycling conditions followed 2 min of PCR initial heat activation at 95 °C, 40 cycles of 10 s at 95 °C for denaturation, and 1 min at 56 °C for combined annealing and extension, and a final melting curve analysis (60–95 °C) at the end. All RT-qPCR analyses were performed using the *elongation factor 1b* (*EF1b; GAWM01010754*) as the reference gene [50] with three technical replicates. Relative expression for *cso-miR-1175* was calculated using ∆Ct [51]. Relative expression for *cso-miR-2944a* and *cso-miR-286a* across time was calculated using the 2-∆∆Ct method [52], with sugar-fed serving as the reference treatment. Expression data between time points were compared statistically using the Kruskal–Wallis test, followed by Dunn’s multiple comparison test. Statistical analyses were performed using GraphPad Prism software version 9.2 (GraphPad Software Inc., La Jolla, CA, USA).

## 3. Results

### 3.1. Small RNA Sequencing Libraries

To determine the *Culicoides* miRNA transcriptome, we collected whole females and midguts at 72 h post blood meal (hpbm) with associated age-matched, sugar-fed controls. In addition, we included data from uninfected *Culicoides* cells (ABADRU-Cs-W8A line). In total, we analyzed ~103.3 million reads obtained from 15 libraries consisting of five sample groups, each with three biological replicates: (1) sugar-fed whole body (SF WB); (2) blood-fed whole body (BF WB); (3) sugar-fed midgut (SF MG); (4) blood-fed midgut (BF MG); and (5) W8A cell culture (Appendix A). After trimming adaptor sequences, ~99.7 million reads remained with a library range of ~16–22 million timed reads (Appendix A) across groups. Most trimmed reads mapped to the *Culicoides* genome across all libraries (97.53–99.62%) (Appendix A). Of the ~98.6 million total mapped reads, an average of ~2.3 million reads (31.93%) from blood-fed midgut tissues, ~2.1 million reads (29.97%) from sugar-fed midgut tissues, ~1 million reads (14.09%) from blood-fed whole females, ~0.9 million reads (13.72%) from sugar-fed whole females, and ~0.4 (6.79%) from W8A cells mapped to miRDeep2-generated *Culicoides* miRNA precursor sequences (Appendix A). In agreement with previous reports in other organisms, *Culicoides* miRNAs composed of a relatively small amount (<32%) of the small ncRNA transcriptome [14,17,53,54].

### 3.2. Annotated miRNAs

Due to the lack of annotated miRNAs in *Culicoides,* we coupled small RNA-Seq and mirDeep2 analyses to determine the midge miRNA catalog. From our small RNA-Seq libraries, we identified 76 miRNAs consisting of 73 potentially orthologous miRNAs and three candidate novel miRNAs (Appendix A). The *Culicoides* miRNA transcriptome also revealed conserved miRNAs across multiple insect species (Appendix A). miRNAs *cso-miR-1175*, *cso-miR-286a*, *cso-miR-2a*, *cso-miR-306*, *cso-miR-316*, and *cso-miR-2944a* were conserved only among Culicidae mosquitoes (*Aedes aegypti*, *Anopheles gambiae*, and *Culex quinquefasciatus*). Both the mature and precursor sequences for *cso-miR-957* were conserved with *Drosophila melanogaster* and identified in other mosquitoes. Our analyses also found eight miRNAs conserved in holometabolous insect species and 20 miRNAs conserved across holometabolous and hemimetabolous insect species. In addition, we found previously reported miRNA clusters in *Culicoides* (Appendix A) such as *miR-1174* and *miR-1175*, the *miR-2a* cluster (*cso-miR-2a*, *cso-miR-2b*, *cso-miR-13*, *cso-miR-2c*, and *cso-miR-71*), and the *miR-309* cluster (*cso-miR-309*, *cso-miR-2944a*, *cso-miR-2944b*, and *cso-miR-286a*) [14,54]. We further identified three candidate miRNAs, *cso-miR-X1*, *cso-miR-X2*, and *cso-miR-X3*, with potential *Culicoides*-specific expression. Of the candidate miRNAs, *cso-miR-X1* lies in a cluster with *cso-miR-283* and *cso-miR-12* (Appendix A), suggesting species-specific changes to the cluster based on synteny over sequence similarity.

### 3.3. Transcriptome Analyses across Blood Meal and Tissue

To focus on miRNAs potentially relevant to midge physiology, only insect samples and miRNAs with at least 200 total read counts were used for expression analyses. Removed miRNAs can be found in Appendix A. Principle component analysis (PCA) showed biological replicates sufficiently grouped, with libraries showing a closer relationship between tissues over blood-feeding treatment (Appendix A). A heatmap was generated from log_10_ transformed reads per million (RPM) (Figure 1; Appendix A), where dendrograms supported PCA data (Figure 1). The heatmap illustrates the range of miRNA expression shared across groups, with *cso-miR-281* and *cso-miR-956* having the highest expression and *cso-miR-965* and *cso-miR-988* having the lowest expression (Figure 1). Our data revealed major expression patterns, including: (1) blood meal-regulated and (2) tissue-enriched expression. These expression profiles were verified through differential gene expression analysis (Table 1 and Appendix A) and RT-qPCR (Figure 2 and Appendix A; see methods for parameters). Due to our tissue analyses, miRNAs enriched in whole-body samples were termed as enriched in non-midgut tissues, which include ovaries, fat body, salivary glands, neuronal, and muscular tissues.

For treatment, blood meal-induced expression was observed in 11 miRNAs (Table 1). Most blood meal-induced miRNAs (91%) were associated with non-midgut tissues, including *cso-miR-133*, *cso-miR-190, cso-miR-2944a*, *cso-miR-957*, and *cso-miR-981*, which have been previously reported to be upregulated after blood feeding in mosquito non-midgut tissues, including the thorax, fat body, and ovaries [15,17,19,22,27]. To further explore the miRNA transcriptome at the tissue level, we determined miRNAs enriched in non-midgut and midgut tissues (Table 1 and Appendix A). Our data showed 24 miRNAs (~32%) enriched in non-midgut tissues, including *cso-miR-1*, *cso-miR-1000*, *cso-miR-124*, *cso-miR-133*, *cso-miR-210*, *cso-miR-285*, *cso-miR-286a*, *cso-miR-2944a*, *cso-miR-927*, and *cso-miR-981*. These miRNAs also displayed non-midgut expression in mosquitoes [15,17,20]. One candidate novel miRNA, *cso-miR-X3*, also displayed non-midgut enriched expression. Midgut-enrichment was observed in 19 miRNAs (~25%), such as *cso-miR-1174*, *cso-miR-1175, cso-miR-281*, and *cso-miR-283*, which is in agreement with previous reports in Culicine mosquitoes [15,20,26] and confirmed for *cso-miR-1175* by RT-qPCR analyses (Appendix A). While *cso-miR-956* was highly expressed in all samples, this miRNA was also upregulated in midgut tissues (Figure 1, Appendix A), which is in agreement with other insects [19,25,54]. Only two midgut-enriched miRNAs changed expression after blood feeding, with *cso-miR-190* and *cso-miR-2796* displaying blood meal-induced and repressed expression, respectively.

Interestingly, we identified multiple members of the *miR-309* cluster, *cso-miR-309, cso-miR-2944b*, *cso-miR-2944a,* and *cso-miR-286a*, but only *miR-2944a* and *cso-miR-286a* underwent differential expression analyses. Both *miR-2944a* and *cso-miR-286a* were enriched in non-midgut tissues and displayed blood meal-induced expression (Table 1 and Appendix A). These findings align with previous reports stating that this cluster was enriched in mosquito ovaries and upregulated after blood feeding [15,17]. Our RT-qPCR data showed *cso-miR-2944a* and *cso-miR-286a* peak expression at 24–48 hpbm (Figure 2), further suggesting that other members of the cluster may share this expression pattern and a role for these miRNAs in *Culicoides* egg development.

### 3.4. miRNAs Potentially Involved in Culicoides–Virus Interactions

To further investigate the miRNAs potentially relevant during viral infection, we performed RNA-Seq and miRDeep2 analyses on previously published small RNA libraries where *Culicoides* KC cells were infected with SBV or BTV [35]. While no additional miRNAs were identified using miRDeep2 from these datasets, we used our *Culicoides* miRNA catalog to check for miRNAs with distinct expression patterns unique to infected cells. Due to limitations regarding the controls associated with available datasets, we focused on the miRNAs with the highest and lowest expression within each dataset, including insect and W8A samples, to identify major trends in miRNA expression (Figure 3, Appendix A). Across all samples, *cso-miR-184* and *cso-miR-8* were among the top ten most highly expressed miRNAs. *cso-bantam* was also within the top ten most highly expressed miRNAs in all samples, except W8A cells. Interestingly, the novel candidate miRNA, *cso-miR-X3*, was among the top ten most highly expressed miRNAs in cell line samples but had moderate expression in midge tissue samples, respectively (Figure 3; Appendix A). While *cso-miR-281* and *cso-miR-956* were among the top ten most highly expressed miRNAs in uninfected tissue samples, these miRNAs were also among the ten least expressed miRNAs in infected cell line samples (Appendix A). While some of these differences in miRNA expression may be cell line-delineated, *cso-miR-33* demonstrated a BTV-specific expression pattern. *cso-miR-33* was among the least expressed miRNAs in many samples but was one of the top ten highest expressed miRNAs in BTV-infected cells. These expression patterns suggest that such miRNAs may have specialized functions related to vector physiology and immunology.

## 4. Discussion

Through small RNA-Seq and miRDeep2 analyses, we determined the miRNA transcriptome of *Culicoides sonorensis* whole females and midgut tissues before and after blood feeding. While these data are the first to annotate *Culicoides* miRNAs, our findings do not imply that the miRNA catalog for *C. sonorensis* is complete. Due to our sampling time, additional miRNAs with differential expression remain to be examined. Nevertheless, we were able to identify 76 miRNAs, including three candidate novel miRNAs for this species, which displayed a variety of expression patterns across blood meal ingestion and tissues at a key viral dissemination time point.

Our data revealed *Culicoides cso-miR-956* was enriched in midgut tissues. In miRBase, *miR-956* is only described in *Drosophila* but has been reported in multiple Dipteran taxa, including Culicomorpha (Culicidae) and Calyptratae muscoids (Sarcophagidae and Calliphoridae) [49,55,56,57,58]. Work on *Drosophila* has alluded to multiple potential functions of *miR-956*, as this miRNA is thought to regulate the circadian rhythm [59] and has been recently shown to have a role in intestinal cell differentiation [60]. While functional studies have yet to be completed outside *Drosophila*, research on *Anopheles* demonstrated that this miRNA was loaded into AGO-1 [17,54], suggesting that this miRNA may share similar physiological relevance across other Diptera, including *Culicoides*. In contrast, *miR-989*, which was shown to be responsible for border cell migration in *Drosohpila* [61] and enriched in mosquito ovarian tissues [15,19,20,62,63], was not identified in our miRNA transcriptome data. Since our samples were collected at 72 hpbm and this miRNA was reported as upregulated after 24 hpbm in *Anopheles* [15], it is possible that *cso-miR-989* had an earlier expression and therefore was overlooked in our analyses. Nevertheless, our experimental design provides a baseline for future studies to probe vector–virus small ncRNA interactions at this infection-relevant time point.

We also identified multiple *Culicoides* miRNA clusters, including members of the *miR-309* cluster, which displayed blood meal-induced expression. In *An. gambiae*, miRNAs in the *miR-309* cluster are enriched in the ovaries and have peak expression at 24 hpbm [15,17]. *Culicoides miR-309* cluster members, *cso-miR-286a* and *cso-miR-2944a,* showed peak expression at 24 hpbm with enriched expression in non-midgut tissues, suggesting other identified members of the cluster, *cso-miR-286a* and *cso-miR-2944b*, may also follow this expression pattern. In *Aedes*, *miR-309* regulates *transcription factor homeobox protein SIX4*, which is required for ovarian follicle growth and maturation [64]. In addition, *miR-309* is also required for the maternal–zygotic transfer of mRNAs in *Drosophila* [65]. These reports suggest that this miRNA cluster is critical for egg development. In addition, we identified a cluster of miRNAs previously reported to be essential for blood digestion and enriched in mosquito midguts [15,20,25,26]: *cso-miR-1175* and *cso-miR-1174*. This cluster displays blood meal-induced expression in mosquitoes with peak expression between 24 and 36 hpbm [66]. As this timeframe was outside our sample time point, additional studies are needed to assess the function of this miRNA cluster in *Culicoides*. Indeed, our data also support the idea that these miRNA clusters are evolutionarily distinct from Culicomorpha, which contain mosquitoes (Culicidae) and midges (Ceratopogonidae). While the *miR-309* cluster is observed in other hematophagous dipterans, stable flies [67] and tsetse flies [55,56], these organisms lack miRNAs *miR-2944a* and *miR-2944b*. Interestingly, *miR-1174* cluster is conserved across mosquito species [68] but not found in current miRNA libraries for other hematophagous Diptera such as *Lutzomyia* [58], *Glossina* [55,56], *Cochliomyia* [57], and *Stomoxys* [67]. Together, these data suggest that the expansion of the *miR-309* cluster and the origin of the *miR-1174* cluster are evolutionarily distinct to Culicomorpha, as mosquitoes (Culicidae) and midges (Ceratopogonidae) share an evolutionary origin for hematophagy that is separate from the independent evolutionary origins of hematophagy in sand flies (Psychodidae), tsetse flies (Glossinidae), screwworms (Calliphoridae), and stable flies (Muscidae) [69].

Culicomorpha contain multiple hematophagous species that vector viral pathogens to humans and animals, with multiple reports of the effects of viral infection on the miRNA transcriptome of well-studied vectors [23,25,70,71]. Currently, only Schnettler et al. have infected *Culicoides* KC cells with SBV at 10 MOI and BTV-1 at 0.2 MOI [35], providing excellent publicly available small RNA libraries for analysis. Due to limitations concerning the availability of uninfected control datasets from the same cell line and differences between infection doses, we focused on identifying the miRNAs with the highest and lowest expression within each dataset. miRNAs *cso-miR-956* and *cso-miR-281* demonstrated contrasting expression patterns between uninfected and infected samples, with downregulated observed in both SBV- and BTV-infected samples. In *Drosophila, miR-956* was also suggested to have antiviral functions by modulating Toll signaling to inhibit a viral replication of *Drosophila* C virus [72], suggesting that this miRNA may have multiple functions. In *Aedes*, *miR-281* expression is also virus-dependent but is upregulated during dengue virus infection [70]. While these miRNAs may change expression due to viral infection, we cannot disregard that these patterns may be due to cell lineage. Interestingly, *cso-miR-33* was the third highest expressed miRNA in BTV-infected cells but among the lowest expressed miRNAs in multiple other samples. In *Culex*, *miR-33* was also upregulated after West Nile virus infection [24], but the function of this miRNA has yet to be determined. Since these miRNAs have unique expression patterns, future experiments are needed to determine their function during viral infection.

Due to the tight association between pathogen transmission, blood feeding, and egg development, understanding midge physiology as it relates to ncRNAs is paramount. To this end, we successfully characterized the miRNA catalog of *Culicoides sonorensis* and identified multiple miRNA expression patterns associated with blood feeding and/or tissue. Our findings also suggest an evolutionary relationship between blood feeding and miRNA expression within Culimorpha. Furthermore, we identified multiple miRNAs with infection-associated expression patterns. Additional studies are needed to further elucidate the miRNA catalog in the midge and explore the expression dynamics correlated with different physiological responses. Overall, our small RNA-Seq data provide the necessary foundation for future small ncRNA work to identify miRNAs associated with midge physiology and immunity.

## Figures and Tables

**Figure 1 insects-14-00611-f001:**
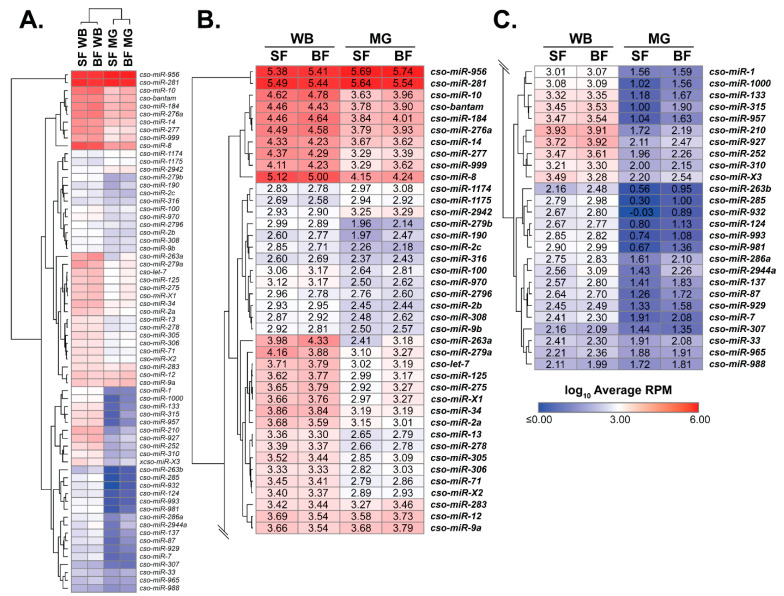
Culicoides miRNA transcriptome. (**A**) The heatmap of *Culicoides* miRNA expression across tissue and blood meal with blue and red denoting minimum and maximum values, respectively. (**B**,**C**) The heatmap is enlarged to view the specific expression values, which are represented as log_10_ transformed average RPM values. Midge samples include whole body (WB) and midgut (MG) tissues, and sugar-fed (SF) and blood-fed (BF) treatments.

**Figure 2 insects-14-00611-f002:**
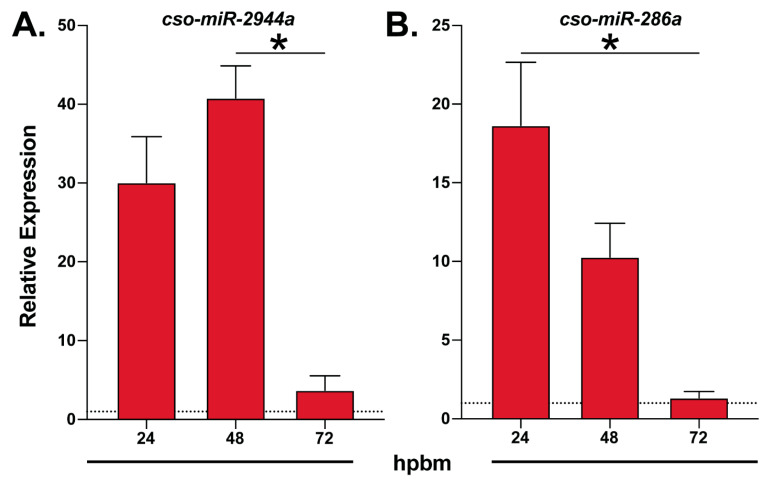
Temporal expression of blood-meal induced midge miRNAs. An illustration of RT-qPCR expression data of (**A**) *cso-miR-2944a* and (**B**) *cso-miR-286a* at multiple time points after blood feeding relative to sugar-fed controls (dotted line). Data are represented as the mean ± SE of three biological replicates. Statistically significant differences are denoted by an asterisk (Kruskal–Wallis, Dunn’s multiple comparison test, *p* < 0.05). hpbm: hours post-blood meal.

**Figure 3 insects-14-00611-f003:**
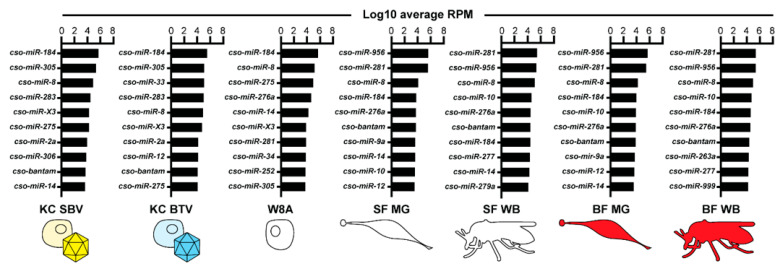
The ten highest expressed miRNAs. An illustration of the top ten highest expressed miRNAs across all datasets. Data are represented as log_10_ transformed average RPM values of two to three biological replicates. Cell line samples include non-infected W8 cells and KC cells infected with either Schmallenberg virus (SBV) or bluetongue virus (BTV). Insect samples include whole body (WB) and midgut (MG) tissues, and sugar-fed (SF) and blood-fed (BF) treatments. RPM: reads per million.

**Table 1 insects-14-00611-t001:** *Culicoides* miRNAs with differential expression patterns across tissue and blood meal.

miRNA	Tissue Enrichment	BM—Expression *
*cso-miR-100*	MG	
*cso-miR-1174*	MG	
*cso-miR-1175*	MG	
*cso-miR-12*	MG	
*cso-miR-190*	MG	+
*cso-miR-2796*	MG	−
*cso-miR-281*	MG	
*cso-miR-283*	MG	
*cso-miR-2942*	MG	
*cso-miR-305*	MG	
*cso-miR-306*	MG	
*cso-miR-308*	MG	
*cso-miR-316*	MG	
*cso-miR-33*	MG	
*cso-miR-956*	MG	
*cso-miR-965*	MG	
*cso-miR-988*	MG	
*cso-miR-9a*	MG	
*cso-miR-9b*	MG	
*cso-miR-1*	n-MG	
*cso-miR-1000*	n-MG	
*cso-miR-124*	n-MG	
*cso-miR-133*	n-MG	+
*cso-miR-137*	n-MG	
*cso-miR-210*	n-MG	
*cso-miR-252*	n-MG	
*cso-miR-263a*	n-MG	+
*cso-miR-263b*	n-MG	+
*cso-miR-277*	n-MG	
*cso-miR-285*	n-MG	+
*cso-miR-286a*	n-MG	+
*cso-miR-2944a*	n-MG	+
*cso-miR-310*	n-MG	
*cso-miR-315*	n-MG	+
*cso-miR-7*	n-MG	
*cso-miR-87*	n-MG	
*cso-miR-927*	n-MG	
*cso-miR-929*	n-MG	
*cso-miR-932*	n-MG	+
*cso-miR-957*	n-MG	+
*cso-miR-981*	n-MG	+
*cso-miR-993*	n-MG	
*cso-miR-X3*	n-MG	

MG: Midgut; n-MG: non-midgut tissues; BM: blood meal. * *Culicoides* miRNAs with a statistically significant ≥ 2-fold increase and decrease in expression after blood feeding are denoted by (+) and (−), respectively.

## Data Availability

Sequencing data from all tissues and treatments have been submitted to the NCBI SRA database under BioProject number PRJNA871270. Sequencing data will also be available through Vectorbase [38] and Ag Data Commons.

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
