# Peer review of "microRNA Expression Dynamics in Culicoides sonorensis Biting Midges Following Blood-Feeding"

_insects, 2023, doi:10.3390/insects14070611_

Round 1

Reviewer 1 Report

The authors present a well written manuscript describing the microRNAs expressed in Culicoides during blood feeding. There are significant gaps in knowledge of the molecular processes that determine the ability of Culicoides species and individuals to act as vectors of economically important viruses of livestock. This manuscript examines the expression of miRNAs through blood feeding in Culicoides, a process that is strongly linked to the transmission of arboviruses, for the first time. For the most part, the manuscript is clear and adds evidence to further our understanding.

 I do question the description of the analysis of publicly available small RNA sets. I think it could be made clearer as to what was analysed, the absence of controls and differences in MOI inherent in the published study, as present it is difficult to see the value of this data as expression may be cell line delineated, as stated in the text and comparison between these data and those derived from insects in the study may not be justified.  The authors acknowledge this, but I think it requires stronger justification for inclusion in an otherwise excellent paper.

Author Response

We thank the reviewer for their comments. As the reviewer stated, there are limitations to the direct comparisons that can be made between Schnettler et al. 2013 and our datasets. We have acknowledged these limitations in the manuscript. In addition, we have not used direct miRNA expression comparisons, and instead, looked for trends across the highest and lowest expressed miRNAs. These data were analyzed for the betterment of the Culicoides vector community, as we still lack large ncRNA datasets for Culicoides sonorensis. The reviewer’s comments have brought to light that our data could be better described and presented to prevent misunderstanding. Overall, we agree that these expression patterns could be cell line-delineated and hope our data provide an avenue for future studies to determine the relevance of miRNAs with expression patterns associated with infected cell lines.

To more clearly present our findings the following sentences have been changed:

  • Line 46: added “potentially”
    • The sentence now reads “Lastly, we further identified miRNAs with expression patterns potentially associated with virus infection…”
  • Line 48: removed “and virus transmission”
    • The sentence now reads “Together, our data provide a foundation for future ncRNA work to untangle the dynamics of gene regulation associated with midge physiology.”
  • Line 348: reworded the sentence
    • The sentence now reads “To further investigate the miRNAs potentially relevant during viral infection, we performed…”
  • Line 352: reworded the sentence
    • The sentence now reads “…we used our Culicoides miRNA catalog to check for miRNAs with distinct expression patterns unique to infected cells.”

Reviewer 2 Report

MiRNAs are well known to regulate gene expression. In blood-feeding midges, they may be involved in egg development as well as in virus/pathogen transmission, three tightly associated physiological processes in blood feeding insects. Authors in this paper successfully characterized the miRNA catalog of a blood-feeding midge, Culicoides sonorensis, including three novel miRNAs for this species, and identified multiple expression patterns, associated with blood feeding in whole females and midgut tissues.  

The experiments seem to be carefully done and the manuscript is written very well. I have only few minor suggestions for improvement:

- line 156 and others: insert a space between number and unit

- line 199 and others: either "against Diptera" or "against dipteran", etc.

- References: Proc Natl Acad Sci USA (!); give all journal names in lowercase letters and not mixed.

Author Response

We thank the reviewer for their comments. We have addressed the comments below and have made changes for better clarification and proper citations. 

The following changes have been made:

  • line 156 and others: insert a space between number and unit
    • The suggested change has been made.
  • line 199 and others: either "against Diptera" or "against dipteran", etc.
    • The suggested change has been made. The sentence now reads “…against Diptera (Aedes aegypti, Anopheles gambiae, Culex quinquefasciatus, Drosophila melanogaster, and Drosophila mojavensis), Lepidoptera (Bombyx mori, Heliconius melpomene, and Manduca sexta), Hymenoptera (Apis mellifera), Coleoptera (Tribolium castaneum), and Hemiptera (Acyrthosiphon pisum) miRNAs…”
  • References: Proc Natl Acad Sci USA (!); give all journal names in lowercase letters and not mixed.
    • All references have been reviewed and the journal names are now in format with Insects

Round 2

Reviewer 1 Report

My concerns have been addressed.